# Emergent Communication with Conversational Repair

**Mitja Nikolaus** [*]
CerCo, CNRS
`mitja.nikolaus@cnrs.fr`

## Abstract

Research on conversation has put emphasis on the importance of a multi-level communication system, in which the interlocutors aim to establish and maintain common ground. In natural conversations, repair mechanisms such as clarification requests are frequently used to improve mutual understanding. Here we explore the effects of conversational repair on languages emerging in signaling games. We extend the basic Lewis signaling game setup with a feedback channel that allows for the transmission of messages backwards from the receiver to the sender. Further, we add noise to the communication channel so that repair mechanisms become necessary for optimal performance.

We find that languages emerging in setups with feedback channel are less compositional. However, the models still achieve a substantially higher generalization performance in conditions with noise, putting to question the role of compositionality for generalization. These findings generalize also to a more realistic case involving a guessing game with naturalistic images.

More broadly speaking, this study provides an important step towards the creation of signaling games that more closely resemble the conditions under which human languages emerged.

## 1 Introduction

Conversation analysis has been describing human conversation as interactions between speaker and listener, in which the interlocutors are using multiple communicative levels to negotiate mutual understanding (Schegloff et al., 1977; Schegloff, 1982; Clark & Schaefer, 1989; Clark, 1996; Pickering & Garrod, 2021). Whenever speakers are verbalizing their communicative intent to a listener, thereby communicating some information, listeners can either acknowledge (explicitly or implicitly) the receipt of this information or initiate a repair routine (e.g., ask for clarification in case they did not understand the speaker correctly).

While conversational repair mechanisms such as clarification requests (also known as other-initiated repairs) have been found to be present in a large range of human languages (Tabensky, 2001; Dingemanse & Enfield, 2015), most recent research on *language evolution* has focused on unidirectional communication channels, thus only allowing information flow from the sender to the receiver, and not backwards. However, for basic other-initiated repair to emerge, a *feedback* information flow from the receiver to the sender is necessary.

In this work, we study the role of conversational repair for the nature of languages emerging in signaling games (Lewis, 1969). We extend a widely-used basic signaling game setup to allow for the flow of feedback messages from the receiver to the sender, thus implementing a bidirectional model of communication.

By studying the languages emerging in this setup, we find that they generalize better to unseen test examples under noisy conditions, while showing a substantially lower degree of compositionality as measured by topographic similarity. We validate this result for a range of different noise levels, messages lengths, and input space sizes.

---

[*]Work performed at Aix-Marseille University.

Finally, we develop a more realistic guessing game setup with naturalistic scenes based on the Guess-What?! dataset (De Vries et al., 2017), in which the receiver needs to discriminate a target object from a set of distractor objects within the same visual scene. Our findings regarding the improved performance under noisy conditions generalize to this more realistic setup.

## 2 RELATED WORK

### 2.1 COMPUTATIONAL MODELING OF EMERGENT COMMUNICATION

Computational models of emergent communication aim to implement aspects of human language evolution using communication games. While early attempts used Bayesian modeling to study the emergence of syntax using the so-called iterated learning model (Kirby & Hurford, 2002; Kirby et al., 2007), more recent approaches are leveraging deep reinforcement learning approaches to scale the models up to more realistic learning scenarios (Lazaridou et al., 2017; Lazaridou & Baroni, 2020; Guo et al., 2022; Chaabouni et al., 2020; Lazaridou et al., 2018; Chaabouni et al., 2022; Rodríguez Luna et al., 2020).

In many studies, emergent communication is studied in a basic Lewis signaling game (Lewis, 1969), which involves a sender and a receiver. The sender is required to communicate some information to the receiver through a communication channel with limited capacity. Most models only consider a unidirectional communication channel, without any possibility for information flow backwards from the receiver to the sender, therefore not allowing for any conversational repair mechanisms to emerge. Exceptions are the game setups in Evtimova et al. (2018); Cao et al. (2018); Graesser et al. (2020), which allow for multi-directional flow of information. However, these studies did not consider communication channels with noise and consequently there exists no pressure for repair mechanisms to emerge. Jorge et al. (2016) analyzes languages emerging in a bidirectional signaling game with noise, but the noise is added to the communication channel in a way that it is not directly detectable by the message receiver.

Here we focus on a bidirectional communication game setup, in which sender messages are replaced by a special noise token with a certain probability. Thereby, the receiver can in principle learn to detect the presence of the noise token and initiate a conversational repair routine.

**Compositionality and Generalization**   A range of computational studies has explored compositionality and generalization in emerging languages. Chaabouni et al. (2020) studies the phenomena in a principled approach and found that agents can succeed to communicate and generalize even to unseen objects without the emerged languages necessarily being compositional according to a range of measures. The authors find that generalization capabilities emerge if the input space is large enough. Rita et al. (2022a) looks into multi-agent game setups and finds that sufficiently heterogeneous populations produce more compositional languages with an increasing number of agents. These results are in line with research on experimental studies with human subjects (e.g., Raviv et al., 2019). Rita et al. (2022b) shows that the commonly used loss can be broken down into an information term and a co-adaptation term, and that controlling for overfitting on the co-adaptation loss increases compositionality and generalization performance. Other studies explore the role of template transfer (Korbak et al., 2021), communication channel capacity (Gupta et al., 2020), or communication over sets of objects (Mu & Goodman, 2021).

In our work we directly compare the generalization performance and compositionality of models with unidirectional communication channel to those with an additional feedback channel.

### 2.2 CONVERSATIONAL REPAIR IN LANGUAGE EVOLUTION

Historically, a large portion of research in linguistics has been dedicated to find universals in the syntax of human languages. While the existence of such a *Universal Grammar* is disputed, more recent trends highlight the possibility to describe universals with respect to the *use* rather than the *structure* of language. For example, it has been argued that certain communicative feedback devices such as other-initiated repair could be universally present in human languages (Dingemanse et al., 2013; 2015; Dingemanse & Enfield, 2015). Such universals of conversation are not explained by innateness, but rather by a selective pressure towards the evolution of common optimised forms that

is exerted by the conversational environments (Dingemanse et al., 2013; Roberts & Mills, 2016). As such mechanisms form major building blocks of human communication, it is important to investigate how they impact the emergence of structure in language (Silva & Roberts, 2016). Healey et al. (2007) analyzes languages emerging between human interlocutors in a graphical language game and finds that repair is key for the emergence of complex symbol systems. Mills & Redeker (2022) suggests that self-repair increases the abstraction of emerging message systems.

Lemon (2022) sketches out a framework for emergent communication with conversational grounding. Agents should be able to detect disagreements and resolve them, in order to maintain a common ground. Targeted feedback signals facilitate the coordination between communication partners. Related computational implementations can be found for example in Steels (1995), where a model for vocabulary formation within conversation that includes simple feedback mechanisms for responses and message acknowledgements is proposed. Other examples include Tria et al. (2012), which focuses on "blending repair", a strategy that exploits the structure of the world to create new words, as well as de Ruiter & Cummins (2012), proposing a bayesian model of communication in which repair sequences are initiated if the entropy of the prior and posterior probability distributions over possible intentions surpass a certain threshold. Finally, van Arkel et al. (2020) compares pragmatic reasoning and other-initiated repair, using bayesian modeling and complexity analysis.

In our work, we explicitly study the role of conversational repair by directly comparing models with and without feedback channel regarding the generalization performance and the compositionality of the emerging languages. Crucially, we leverage deep-learning based models that scale to more realistic input, instead of only small-scale toy language game setups.

## 3 METHODS

### 3.1 BASIC SIGNALING GAME

We implement a signaling game (Lewis, 1969) following common practices in the literature (Kottur et al., 2017; Lazaridou et al., 2018; Chaabouni et al., 2020; Rita et al., 2022b). In the following, we will describe the details for the baseline used in all experiments.

Two agents communicate using symbols in a discrimination game. A sender agent $S$ is provided with an input object $o_i$, sends a message token $m \in X$ using discrete symbols to the receiver agent $R$. The vocabulary of possible tokens is denoted as $X$. The receiver needs to discriminate the target object from a set of distractor objects $O$ by using the information provided in the message $M$. The input objects are defined by a number of attributes $A$ each with possible values $V$. An object is encoded using a concatenation of one-hot encodings for each attribute, i.e. the input dimensionality is $|A| \cdot |V|$. The capacity of the communication channel is defined by the number of symbols in the vocabulary $|X|$ and the message length $|M|$.

Both sender and receiver are implemented as gated Recurrent Neural Networks (RNNs) using single-layer GRUs with layer normalization (Ba et al., 2016). In the basic setup, the number of distractor objects (including the target) $|O|$ is set to 2. The parameters $\theta_R$ of the receiver are optimized using a cross-entropy loss:

$$L_{receiver}(\theta_R) = -log(\pi_{\theta_R}(o_i|O, M) \tag{1}$$

where $\pi_{\theta_R}$ is the current policy of the receiver.

In parallel to the receiver, the sender agent is trained using REINFORCE (Williams, 1992):

$$L_{sender}(\theta_S) = -\sum_{t=0}^{|M|} r \cdot log(\pi_{\theta_S}(m_t|o_i, m_{t-1})) \tag{2}$$

where $\pi_{\theta_S}$ is the current policy of the sender, $m_t$ is the message token at time step $t$, and $r$ is the reward ($r = 1$ if the receiver chooses the correct object from the set of distractor objects and $r = 0$ otherwise). We further use a running mean baseline to reduce the variance of the gradients as well as entropy regularization to encourage exploration. At training time, the messages from the speaker are sampled from the current policy, at test time we employ greedy decoding.

We split the set of all possible objects into a training set (90%) and a test set (10%). Further hyperparameters and implementation details can be found in Appendix A.1. The source code of the mod-

els and all experiments is publicly available at `https://github.com/mitjanikolaus/emergent_communication`.

## 3.2 BASIC SIGNALING GAME WITH NOISE AND FEEDBACK

To explore the effects of feedback, we make two adjustments to the baseline model described in the preceding section. First, we introduce noise to the communication channel: With a probability of $p_{noise}$, each token in the message $M$ is replaced with a special noise token.[1] $M'$ denotes the message after manipulation with the noise. Secondly, we allow the receiver RNN to generate feedback messages. At each timestep, the receiver RNN consumes the sender message token and produces a feedback token $n \in Y$. The sender RNN consumes this feedback token in addition to its last turn's output (both tokens are embedded and afterwards concatenated).

The loss functions for the agents with feedback are as follows:

$$L_{receiver\_fb}(\theta_R) = -log(\pi_{\theta_R}(o_i|O, M', N)) \tag{3}$$

$$L_{sender\_fb}(\theta_S) = -\sum_{t=0}^{|M|} r \cdot log(\pi_{\theta_S}(m_t|o_i, m_{t-1}, n_{t-1})) \tag{4}$$

We set $|Y|$ to 2, i.e. the receiver only produces binary feedback. This allows a receiver agent to use the feedback channel for example to send acknowledgements or open clarification requests (Dingemanse & Enfield, 2015). We leave the study of larger feedback channels for future work. The architecture of the model with feedback channel is displayed in Figure 1.

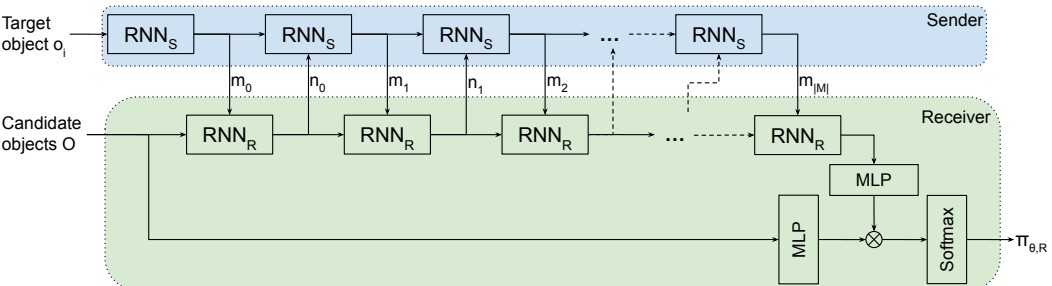

Figure 1: Architecture of signaling game with feedback channel. Both the Sender RNN ($RNN_S$) and Receiver RNN ($RNN_R$) are unrolled in time.

## 3.3 GUESSWHAT SIGNALING GAME

In order to test whether the results observed on the toy signaling game setup generalize to more realistic game setups, we develop another game setup in which agents communicate about objects in naturalistic images. In this game, the receiver has to discriminate a target object from a set of distractor objects that are all present in the same visual scene. This task resembles a common communicative task, in which a speaker is trying to refer to a single object within a visual scene.[2]

The proposed game is based on the GuessWhat?! dataset (De Vries et al., 2017), which was initially designed to create models of grounded task-oriented dialog. Here, we only use the annotated image data, which consists of images annotated with objects and their corresponding bounding boxes (Lin et al., 2014). For each image, we select one of the objects as the input object $o_i$ and use the remaining objects as distractor objects.[3] The remaining task procedure as well as the model implementation

---

[1]See Section 4.1.4 for a discussion of alternative noise implementations.

[2]Related work has proposed to study emergent communication using images from ImageNet (Russakovsky et al., 2015). Here, we propose a task which relies on discriminating objects *within* the same visual scene as opposed to different images, which is arguably harder and at the same time close to communication problems that humans are usually facing: Referring to an object in the shared visual environment.

[3]We constrain the maximum number of distractor objects to 10. If there are more objects available, we randomly sample a subset of 10 objects.

are identical to the basic signaling game (cf. Section 3.1). Two example images are shown in Appendix A.2.

Following the procedure described in De Vries et al. (2017), we select all objects with bounding boxes of a minimal size (area $\geq 500px^2$). We further discard all images that contain only one object. For each object, we extract features from the corresponding bounding box using Vision Transformer (vit-b-16; Dosovitskiy et al., 2020), which yields 768 dimensional vectors. We keep the original train and validation splits as defined in CoCo (Lin et al., 2014). In total, there are 70,702 images (385,961 objects; 5.5 per image on average) in the training split and 8,460 (45,541 objects; 5.4 per image on average) in the validation split (which we use as test set).

## 3.4 EVALUATION

For each setting, we start 3 different runs with varying random seed and report the mean and 95% confidence intervals for all metrics unless stated otherwise. We evaluate the models by measuring accuracy on a held-out test split (test_acc). We further report test accuracy in a separate forward pass for which the channel noise is disabled (test_acc_no_noise). This allows us to investigate how models are performing under optimal conditions even if they were trained with exposure to noise. Finally, we measure the compositionality of the emerged languages using topographic similarity (topsim; Brighton & Kirby, 2006), as it is common practice in the language emergence literature (Lazaridou et al., 2018; Chaabouni et al., 2020; Li & Bowling, 2019). For fair comparison, the compositionality metric is calculated in the separate forward pass during which the channel noise is disabled.

# 4 RESULTS

## 4.1 BASIC SIGNALING GAME

### 4.1.1 EFFECT OF NOISE

We start by investigating the case of $(|A|, |V|) = (4, 4)$ for increasing amount of noise: $p_{noise} \in \{0, 0.1, 0.3, 0.5, 0.7, 0.9\}$. To ensure convergence of the agents, following the results of Chaabouni et al. (2020), we employ them with a large enough channel capacity: A vocabulary size $|X|$ of 2 and a message length $|M|$ of 10.[4]

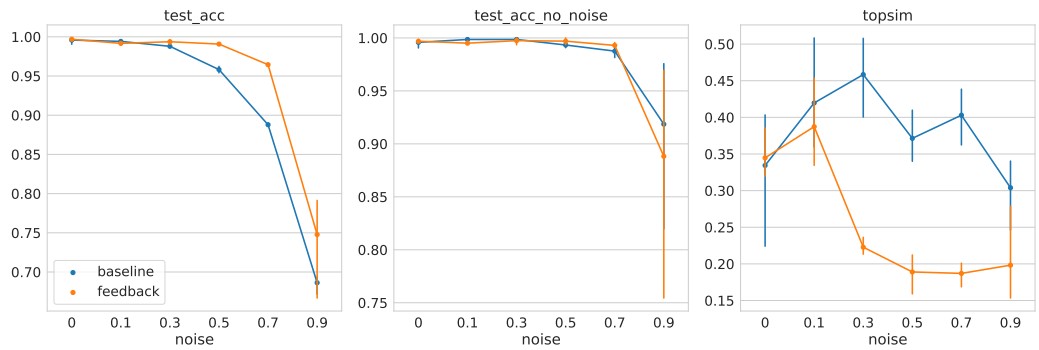

Figure 2: Generalization performance and compositionality scores for models as a function of channel noise $p_{noise}$.

As a first sanity check, we observe that without noise, both models perform optimally (test_acc $\approx$ 1). When comparing the test accuracy in settings with noise, we observe that for all settings the models with feedback outperform the baseline models. This suggests that the feedback channel allows the models to repair the communication under noisy conditions. Additionally, we find that higher noise increases the performance advantage of a feedback channel up to a noise level of $p_{noise} = 0.7$. At

---

[4]In the case of $(|A|, |V|) = (4, 4)$ the input space is $|V|^{|A|} = 4^4 = 256$. In that way the channel capacity is sufficiently larger than the input space: $|X|^{|M|} = 2^{10} = 1024 \gg 256$.

$p_{noise} = 0.9$ the advantage decreases again and the model convergence becomes more unstable (as indicated by the increased variability of performance between runs).

Under optimal conditions, if the channel noise is removed, both models perform approximately on par, suggesting that while the feedback models can repair communication under noise, this does not harm their performance when noise is absent.

While the test accuracy of feedback models under noise is clearly superior, we observe a substantial drop in the topsim score for these models. This suggests that while the feedback allows the models to *increase* test accuracy in conditions with noise, this is coinciding with an *decrease* in compositionality (as measured by the topsim score). While Chaabouni et al. (2020) already observed that compositionality is not necessary to achieve generalization, here we even observe an opposing trend.

**Analysis of Feedback Messages**   In order to gain a better understanding of how the models employ the feedback channel to repair the communication, we analyze the messages of a converged model for the case $p_{noise} = 0.5$.[5]

To this end, we record the messages sent by the sender as well as the feedback messages sent by the receiver for the test set. Then we calculate the correlation (Matthew's Correlation Coefficient; Matthews, 1975) of receiver messages with (1) the presence of noise in the sender messages, (2) the sender messages (excluding messages that contain noise), as well as (3) the one-hot encodings of the two input objects. Figure 3 visualizes the correlations using heatmaps.

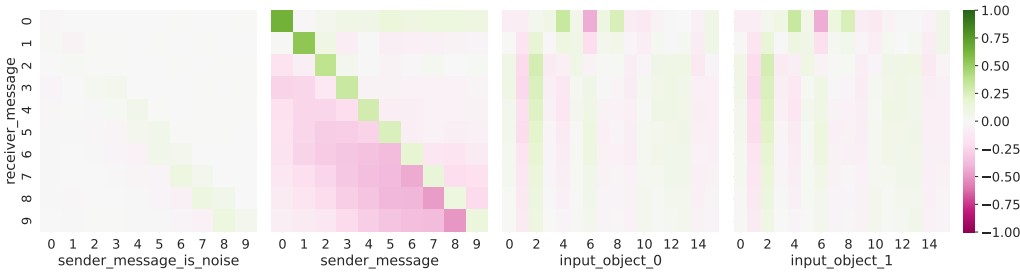

Figure 3: Matthew's Correlation Coefficient between receiver messages and the presence of noise, the sender messages, and the one-hot encodings of the two input objects. The messages are recorded while the agents are playing the signaling game on the test set.

When observing the response patterns we find that the feedback message tokens do not depend on the presence of a noise token in the previous turn (all correlation coefficients are close to 0 in the leftmost graph). This indicates that the feedback tokens are not used as open clarification requests, i.e. they are not simply signaling the presence of noise back to the sender.

The second graph shows that there is a however a positive correlation between the sender messages and receiver messages in the subsequent turn. Following a 1 sent by the sender, the receiver usually responds with 1 and vice versa. In this way, the feedback messages can function as an acknowledgement, signaling the received message back to the sender. For later messages (after message 5 approximately), we find a negative correlation that is slightly delayed.

Finally, we find that there are also substantial correlations between the properties of the candidate objects (target and distractor) and the receiver messages. This hints that the feedback messages *also* serve to communicate certain aspects of the candidate objects to the sender (who does not have access to both objects). In this way, sender and receiver can be co-constructing meaning during the course of the interaction.

Understanding the exact mechanisms of the feedback messages remains challenging, as the models could create any arbitrary messaging code. Still, we would like to estimate to which degree the models actually develop an efficient code to solve the signaling game. We implement an additional setup in which the receiver model is encouraged (using an additional loss term) to only signal the

---

[5]We also analyze the messages of 2 other runs with different seeds and observe highly similar patterns.

presence of noise back to the listener. The details of this setup as well as result graphs can be found in Appendix A.3. We find that while in this case the receivers indeed signal the presence of noise, the generalization performance lacks behind that of models who develop their own feedback messaging code (but is still better than baseline performance without any feedback). The best performing models leverage the feedback message channel to exchange information more efficiently than models using the feedback channel for simple open clarification requests.

### 4.1.2 EFFECT OF INPUT SPACE

To ensure that the observed effects are not only a phenomenon of the specific input space, we experiment with multiple other configurations of larger and smaller input spaces. We keep the noise ratio at $p_{noise} = 0.5$ and vary the number of input attributes $|A|$ and values $|V|$: $(|A|, |V|) \in \{(2, 10), (4, 4), (3, 10), (2, 100), (2, 1000), (10, 1000)\}$.

The results are depicted in Figure 4. We find that for all tested configurations, the feedback channel alleviates the detrimental effects of noise. The largest effects are observed for very small input sizes $(|A|, |V|) = (2, 10)$ or very large ones $(|A|, |V|) = (10, 1000)$. Notably, the input space is even surpassing the channel capacity in the three larger input space settings. In line with the findings of the previous section, we also observe a decrease in topsim scores for most settings. Also, the models' generalization performances are comparable if the channel noise is removed.

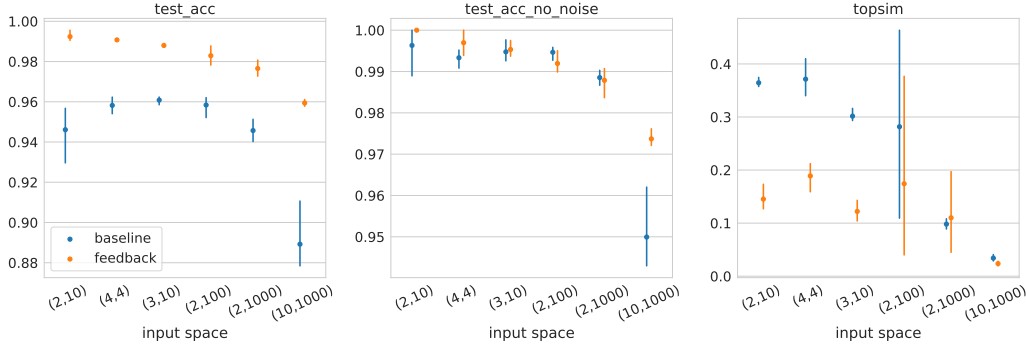

Figure 4: Results for different input space dimensions.

### 4.1.3 EFFECT OF MESSAGE LENGTH

Another important hyperparameter of the game setup is the message length of the communication channel. Here, we investigate the influence of this parameter on the performance advantage of a feedback channel.

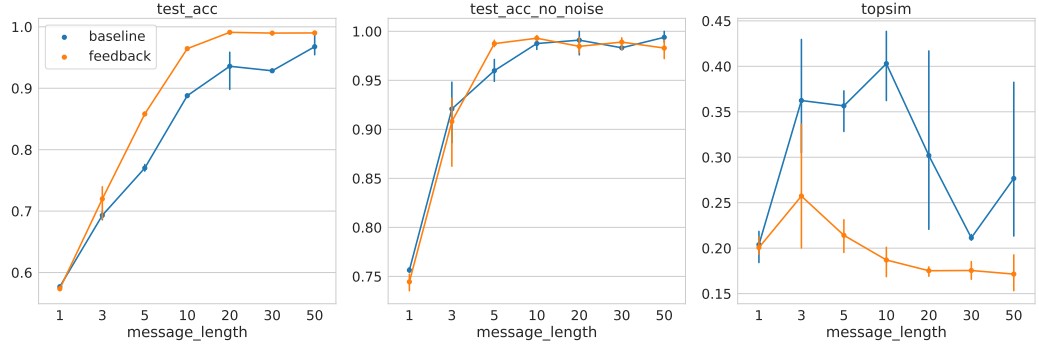

Figure 5: Results as a function of message length $|M|$.

We set $p_{noise} = 0.7$ and vary the message length: $|M| \in \{1, 3, 5, 10, 20, 30, 50\}$. As shown in Figure 5, we find that starting from $|M| = 5$, a performance advantage for the models with feedback emerges. The advantage increases until a length of 30, afterwards the gap between the performance of two model types decreases again. With a sufficiently high message length, the sender can simply repeat each message multiple times to increase chances of successful transmission without the need for any receiver feedback. When comparing the conditions $|M| = 10$ and $|M| = 20$, we find that models with an additional feedback channel and $|M| = 10$ even outperform models with a unidirectional message channel that is double in size ($|M| = 20$). This suggests that in this configuration it is more efficient to allow models for feedback communication than to increase the capacity of the unidirectional message channel.

### 4.1.4 EFFECT OF NOISE IMPLEMENTATION

In our basic game setup the noise is implemented using a special token and is therefore simply detectable by the receiver agent. This relates to phenomena such as a listener not understanding a syllable or word because of some increased background noise. In order to model other phenomena, such as *misunderstandings*, the noise on the channel can be implemented as a random permutation of the message token with another token from the vocabulary. In this case, the presence of noise is not directly detectable by the listener and therefore more negotiation might be necessary in order to obtain a common ground. We therefore expect a lower generalization performance with this implementation of noise.

We run the experiments described in Section 4.1.1 with this alternative implementation of noise. The results are shown in Appendix A.4. We find that for this kind of noise, the generalization performance drops more substantially with increasing noise level (e.g. mean test_acc of 0.70 vs. 0.89 for $p_{noise} = 0.7$), validating our hypothesis that this kind of noise is more challenging for communication. However, we still observe that feedback partially alleviates the effects of noise: The models with feedback outperform the baseline models. The compositionality of languages as measured by topsim is again lower for the models with feedback.

### 4.2 GUESSWHAT SIGNALING GAME

Based on the GuessWhat signaling game described in Section 3.3, we perform a set of experiments to investigate whether the findings on the basic signaling game also hold on more realistic communication game setups with naturalistic images.

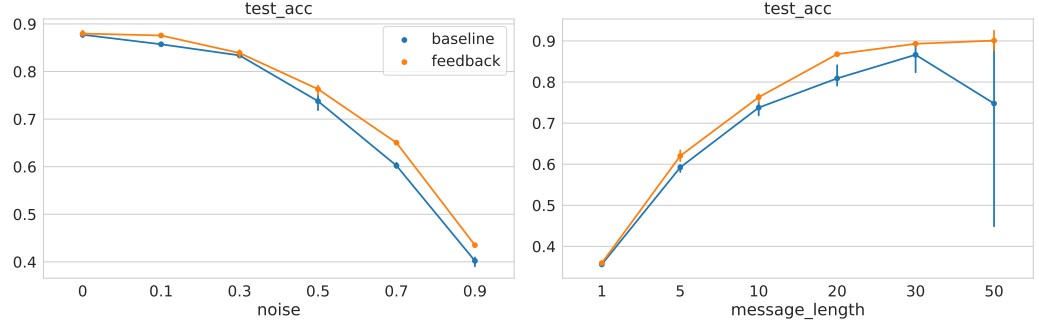

Figure 6: Generalization performance for models in the GuessWhat signaling game as a function of channel noise $p_{noise}$ (left) and message length $|M|$ (right).

We initially keep the same channel capacity as in the basic signaling game setup, a vocabulary size $|X|$ of 2 and a message length $|M|$ of 10. The left plot in Figure 6 shows the effect of increasing noise on models with and without feedback channel. In line with the previous findings, we find that the feedback channel alleviates the effects of noise, with a peak in performance difference that is again around $p_{noise} = 0.7$.

Regarding the role of message length, the right plot in Figure 6 shows that the performance advantage increases with increasing $|M|$ (with a fixed channel noise of $p_{noise} = 0.5$). In contrast to the findings on the basic signaling game, this advantage does not decrease for the largest message length ($|M| = 50$). When evaluating the generalization capabilities without noise, both model types perform comparably (see Appendix A.5).

## 5 DISCUSSION AND CONCLUSION

The findings of this work suggest that in signaling games with noisy conditions, a superior performance can be achieved when models are allowed to send feedback messages backwards from the receiver to the sender. While this increases the generalization performance of the models, the compositionality of the emerged languages decreases.

This drop in compositionality might be explained by multiple factors. First, as already shown in Chaabouni et al. (2020), there is not always a direct link between compositionality and generalization performance. Secondly, natural languages are not perfectly compositional either, in many cases meaning is dependent on context (Goldberg, 2015). When allowing for a bidirectional information flow between sender and receiver, it is possible that both agents are *jointly co-constructing* mutual understanding and thereby creating contextualized meanings. Consequently, the sender messages become less compositional and more context-dependent (see also Section 4.1.1).[6] Recently, Korbak et al. (2020); Conklin & Smith (2023) also highlighted the limitations of topsim as a measure of compositionality in emergent communication, to which our results add additional evidence.[7]

Lemon (2022) pointed to a lack of vision-and-language datasets that explicitly require conversational grounding in additional to symbol (visual) grounding. In this work we designed a simple referential signaling game that allows for the study of conversational repair in the context of a referential game within naturalistic scenes. In line with the findings from the basic signaling game, we find that a feedback channel allows models to improve their generalization performance under noise. With the development of models for an efficient generation of clarifying questions in dialog being an open challenge (Kiseleva et al., 2022), the proposed setup allows for the study of the emergence of crucial mechanisms for successful dialog, such as basic communicative grounding acts (Clark & Schaefer, 1989; Clark, 1996).

So far, this work only investigated setups with binary message and feedback channels. To study the emergence of more advanced repair mechanisms such as restricted requests or restricted offers as opposed to open clarification requests (Dingemanse & Enfield, 2015), the capacity of the message channel should be increased in subsequent works.

We experimented with two alternative implementations of noise (cf. Section 4.1.4), but even further setups should be investigated in the future and might trigger the emergence of more advanced repair mechanisms. This includes for example *combining* the two proposed noise implementations (special noise token for modeling non-understanding, and token permutations for modeling misunderstanding) within a single model, as well as non-uniform distributions of noise. Relatedly, we currently do not add any noise on the feedback messages from the receiver. While this design choice was taken to study the emergence of basic conversational repair, it is not realistic and will need to be adapted in the future to perform more extensive experiments on *nested* clarification requests (van de Braak et al., 2021). Other axes of future work could extend the model to explore the emergence of a preference for self-repair over other-initiated repair, which is typically found in human conversation (Schegloff et al., 1977).

As indicated from these numerous opportunities for future work, the current work contributes another important step to the ongoing efforts on closing the gap between signaling games and realistic models of language evolution (Chaabouni et al., 2019; Rita et al., 2020; Galke et al., 2022).

---

[6]Kottur et al. (2017) also observe that agents exploit bidirectional communication channels to create non-compositional languages. They counteract by limiting the vocabulary size and removing one agent's memory at every timestep, which prevents messages from being context-dependent.

[7]LaCroix (2019) questions compositionality as a target for language evolution research more generally. The author argues that focus should instead be put on *reflexivity*, as it is more consistent with a gradualist approach to language origins. Future work is required to operationalize measures of reflexivity and apply them to computational emergent communication experiments.

ACKNOWLEDGMENTS

Many thanks to Lukas Galke for fruitful and in-depth discussions related to this work. Further, this work was substantially improved thanks to the reviewers who provided very constructive feedback.

This work, carried out within the Labex BLRI (ANR-11LABX-0036) and the Institut Convergence ILCB (ANR-16CONV-0002), has benefited from support from the French government, managed by the French National Agency for Research (ANR) and the Excellence Initiative of Aix-Marseille University (A*MIDEX).

The project leading to this publication has received funding from Excellence Initiative of Aix-Marseille - A*MIDEX (Archimedes Institute AMX-19-IET-009), a French "Investissements d'Avenir" Programme.

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

## A  APPENDIX

### A.1  HYPERPARAMETERS

Hyperparameters were configured as indicated in Table 1, unless stated otherwise.

| | |
|---|---|
| optimizer | Adam |
| initial_learning_rate | 0.001 |
| batch_size | 1000 |
| gradient_clipping | 1 |
| message_length | 10 |
| vocab_size | 2 |
| sender_embedding_size | 16 |
| sender_hidden_dim | 128 |
| sender_entropy_coefficient | 0.01 |
| receiver_embedding_size | 16 |
| receiver_hidden_dim | 128 |
| receiver_entropy_coefficient | 0.01 |

Table 1: Hyperparameter settings.

### A.2  GUESSWHAT SIGNALING GAME EXAMPLES

Figure 7 shows two examples for the images used in the GuessWhat signaling game as described in Section 3.3. The receiver agent needs to discriminate the target object (for example the gray parrot in the left figure) from the other objects in the scene (the two other parrots). The task becomes challenging for cases in which the target object is highly similar to some of the distractor objects (for example, discriminating on of the sheep from the others in the right image).

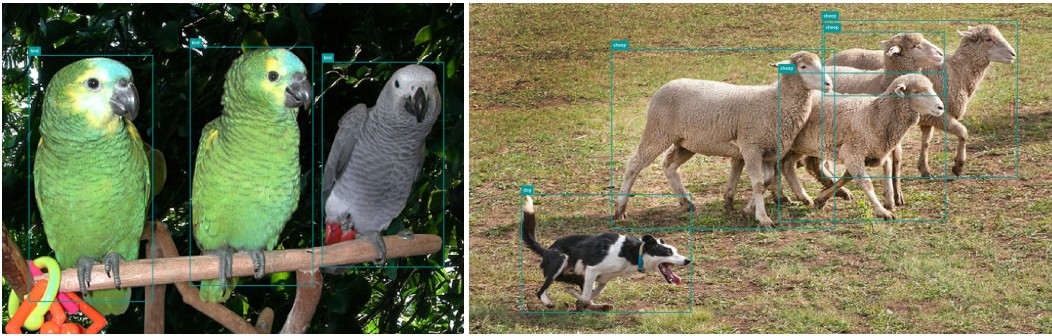

Figure 7: Examples for images used in the GuessWhat signaling game. Candidate objects are highlighted with the colored bounding boxes.

### A.3  RESULTS WITH ADDITIONAL LOSS TERM

We train models with an additional loss term on the receiver side, that is encouraging the receiver messages to signal the presence of noise in the sender messages. The loss is defined as cross-entropy between the receiver message token logits and the presence of noise in the preceding sender message token (1 if noise is present and 0 otherwise). The performance of this model is displayed in Figure 8.

Additionally, we plot an analysis of the feedback messages (see also Section 4.1.1) for a model with $p_{noise} = 0.5$ in Figure 9. This shows clearly that the model is signaling the presence of noise, and (almost) no other information.

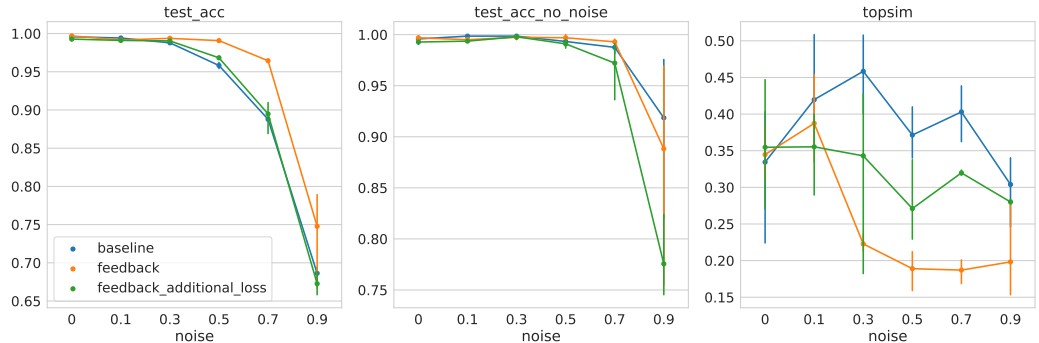

Figure 8: Generalization performance and compositionality scores for models as a function of channel noise $p_{noise}$, including model with additional loss term.

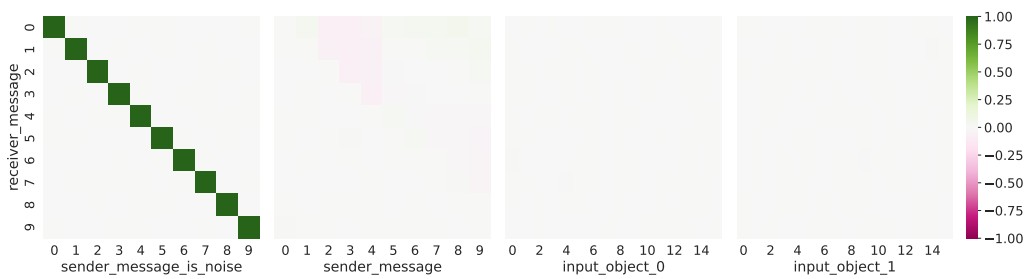

Figure 9: Matthew's Correlation Coefficient between receiver messages and the presence of noise, the sender messages, and the one-hot encodings of the two input objects.

### A.4   RESULTS WITH ALTERNATIVE NOISE IMPLEMENTATION

Figure 10 presents the effect of the alternative noise implementation using message token permutation instead of a special noise token. Figure 11 presents an analysis of the feedback messages for this setup.

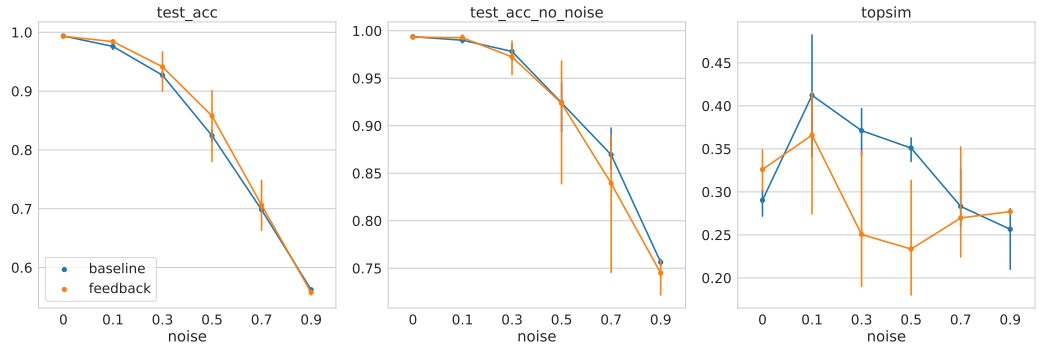

Figure 10: Generalization performance and compositionality scores for models as a function of alternative channel noise $p_{noise}$.

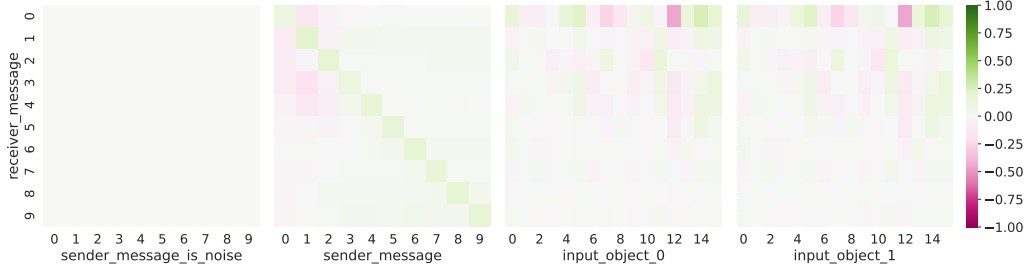

Figure 11: Matthew's Correlation Coefficient between receiver messages and the presence of noise, the sender messages, and the one-hot encodings of the two input objects for a model with alternative channel noise of $p_{noise} = 0.5$. The correlation with the presence of noise is always 0, as there is no explicit noise token in this setup.

## A.5 ADDITIONAL RESULTS FOR GUESSWHAT SIGNALING GAME

Figure 12 presents the results on the GuessWhat signaling game when evaluated without channel noise.

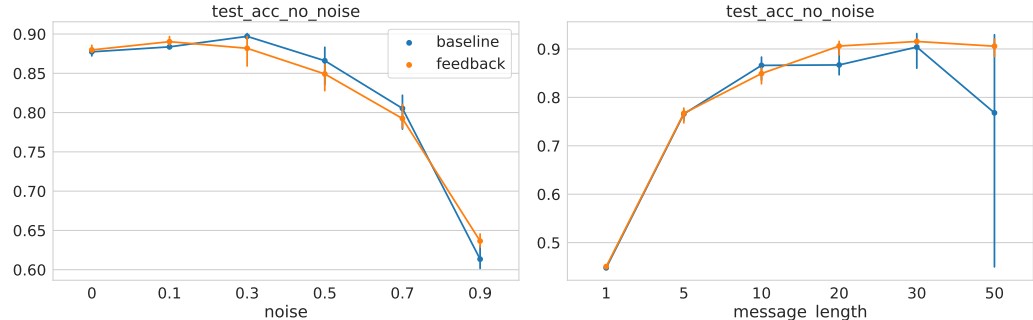

Figure 12: Generalization performance without noise for models in the GuessWhat signaling game as a function of channel noise during training $p_{noise}$ (left) and message length $|M|$ (right).

