# OpenReview forum: "Emergent Communication with Conversational Repair"
_ICLR.cc/2024/Conference — ICLR 2024 poster_

### Official Review · Reviewer_3F5Q · 2023-10-28

**Soundness:** 4 excellent
**Presentation:** 4 excellent
**Contribution:** 3 good
**Rating:** 8
**Confidence:** 4

**Summary:**

This paper introduces a basic notion of communication failure and opportunity
for recovery into the standard emergent communication signalling game.
Communication failure occurs when one of the sender's tokens is replaced with
a special noise token, and the listener is provided with a binary feedback
channel that could be used to indicate the need for recovery.  The empirical
results show that the binary feedback channel provides an increase in
performance in the presence of such noise but is not necessary in the noiseless
cases.  This effect is robust across different hyperparameters as well as with
both symbolic and image embedding-based observations.

**Strengths:**

The paper is coherent: it has a simple, well-defined scope that is explained
well, pursued with reasonable methods, and has empirical data with supports the
main contribution with appropriate ablations.  While the contributions are not
extensive or revolutionary, I do not see this as a problem because the
contributions are sufficient and the quality of the paper as a whole makes
a good building block in the field's body of literature.

**Weaknesses:**

I do not see any major weaknesses in the paper as a whole.  Any specific areas
that could be improved upon are mentioned in "Questions".  What puts my rating
at an 8 instead of 10 is primarily the modest significance of the
contributions.

A minor weakness is that there is no robust/empirical explanation for the cause
of the drop in topographic similarity in the feedback-enabled agents.

**Questions:**

- Page 2:
    - Section 2.2: [Travis LaCroix's paper](https://www.semanticscholar.org/paper/Biology-and-Compositionality%3A-Empirical-for-LaCroix/6422d5e83caec99487936035cfbb2b0d18f2a76d) on reflexivity could be relevant
- Page 5:
    - "instable" -> "unstable"
    - It would be better to use 95% confidence intervals instead of raw standard error
    - Discussion: "jointly co-constructing" seems pretty relevant to Kottur et al. (2017), maybe a sentence connecting the two works would be appropriate
    - "lack vision-and-language" -> "lack of vision-and-language"?
    - Would it be possible to give a sense of what the correlation between the
      presence of the noise token and the feedback request from the receiver
      is?  It's not necessary for all of the experiments, maybe just the
      initial basic ones.

---

> ### Author Response · Authors · 2023-11-22
>
> Thanks a lot for your thorough review!
>
> Regarding the drop in topsim scores:
> - As mentioned by Reviewer p6xc, the sender’s messages in the feedback case depend on both the input object and the receiver’s feedback. In our response to this review, we investigated whether calculating topsim scores that also take into account the receivers messages (by concatenating the sender messages and the receiver messages) leads to improved topsim scores. This is however not the case: These scores are even lower than the original topsim scores (model with $p_{noise}=0.7$, $(|A|, |V|) =  (4, 4)$: topsim_sender_receiver=0.15+-0.02 | topsim=0.19+-0.02). One possible confound with this approach however is that the dimensionality of the meaning space is increased in the feedback case, which makes the topsim scores not directly comparable anymore to the scores in the non-feedback case.
> - We believe that topographic similarity is actually not an appropriate measure for compositionality in this game. As you also mentioned in your response to Reviewer R7xc (“Responding to above weaknesses”), there is increasing evidence for the inadequateness of this measure. Most recently, [1] proposed 4 measures of linguistic variation, which are more appropriate as they correlate (negatively) with generalization performance. One possible explanation for the drop in topographic similarity is therefore that it is simply not the adequate measure. Future work should explore whether other measures such as linguistic variation are better at reflecting the compositionality/ regularity of emerging languages. Most related to the work at hand, new metrics that take into account the feedback messages of the listener will be required.
>
> Regarding Travis LaCroix's paper on reflexivity
> - Thanks, this paper offers new interesting perspectives. We integrated it into the discussion of the updated manuscript. The lack of compositionality found in our study could indeed provide further evidence that other aspects of human languages, such as reflexivity, should be taken into account.
>
> Regarding the discussion: "jointly co-constructing" seems pretty relevant to Kottur et al. (2017), maybe a sentence connecting the two works would be appropriate
> - This is indeed very related, we added a footnote (Footnote 7) to the updated version of the manuscript.
>
>
> Would it be possible to give a sense of what the correlation between the presence of the noise token and the feedback request from the receiver is? It's not necessary for all of the experiments, maybe just the initial basic ones.
> - We made efforts to analyze the semantics of the feedback messages of the receiver agents in the updated version of the manuscript. (See also response to Reviewer p6xc and Section 4.1.1. in the updated paper)
>
>
> Other points/typos:
> - "instable" -> "unstable" and "lack vision-and-language" -> "lack of vision-and-language": Done
>
> - It would be better to use 95% confidence intervals instead of raw standard error: Done
>
>
> [1] Conclin & Smith 2023 Compositionality with Variation Reliably Emerges between Neural Networks

---

### Official Review · Reviewer_p6xc · 2023-10-30

**Soundness:** 3 good
**Presentation:** 3 good
**Contribution:** 3 good
**Rating:** 6
**Confidence:** 3

**Summary:**

This paper introduced a new Lweis signaling game setup with a noisy communication channel for emergent communication. Through experiments in logical and pixel-level input, they showed that setups with the receiver’s feedback can achieve better generalization performance in a noisy environment. By reporting the topographical similarity, it also points out that a better generation does not usually mean a higher compositionality.

**Strengths:**

1. This paper considers a novel setup with multi-step communication with the receiver’s feedback. It changes the original unidirectional communication to bidirectional communication, which can better resemble realistic human communication scenarios.

2. The experiments designed multiple evaluation groups to test how a feedback channel helps alleviate the effects of noise, influence generalization, and compostionality. The experiments introducing realistic image referential tasks with objects in the same visual background also look interesting. The performance contrast between compositionality and generalizability and the potential cause of the receiver’s feedback is worth further investigation.

**Weaknesses:**

The current experiments only show that communication with the receiver’s feedback will generalize better in a noisy environment. It would be more comprehensive to understand this feedback behavior with further analysis:

    a. The semantics of the feedback token: for example, is it more related to interaction regulation (continue to talk, no need to talk) or the attribute of the objects (clarification on some attributes)?

    b. The messages updated by the sender: based on the receiver’s feedback, how the sender’s messages vary across different time steps.

    c. Through multiple iterations, will the receiver provide less feedback or less informative feedback while they gradually build their common ground?

    d. Will the emerged languages with additional feedback make the sender’s messages of different objects more separatable?

**Questions:**

1. How does the game end? Will they communicate a fixed number of time steps?
2. How does the receiver generate the feedback token, through another MLP layer besides the one used in the final target selection?
3. The drops in the compositionality are interesting. Since the sender’s message now depends on both the symbolic input and the receiver’s feedback, the semantic spaces could be disturbed. Have you tried to let the sender and the receiver share the same vocabulary?

---

> ### Author Response · Authors · 2023-11-22
>
> Many thanks for your review!
>
> Regarding the weaknesses:
> - a: We made efforts to try and understand the semantics of the feedback token in the latest updated version of the manuscript (Section 4.1.1). In short, the feedback messages are most likely used for both (1) a form of message acknowledgement, signaling the received message back to the sender and (2) to signal some information about the input objects back to the sender (who does not have access to both objects). The exact semantics of the feedback signals are not trivially decodable, but we demonstrate through additional experiments that the models come up with a messaging protocol that is more efficient than a simple signal indicating the presence of noise back to the sender.
> - b: This is very related to the previous question, and is depending on a better understanding of the feedback signal of the receiver.
> - c: This is a very important point. The nature of feedback during the training iterations is surely developing, however, we do not have a notion of what is “more” or “less” feedback as the feedback signal of the receiver is not directly interpretable (also the receiver is always sending a feedback message, there is currently no option to stay silent implemented). More generally, in the current setup there is always noise on the communication channel, which means that some form of feedback will stay useful even for 2 agents who already established a common ground. Future work could additionally explore agent populations, in which some agents speak more often to some than to others, in order to explore effects of pre-established common ground between speakers. (see also [1])
> - d: More distinct messages for different objects allow for a higher discrimination performance. As we indeed observe a higher test accuracy, we conclude that most likely these senders produce more separatable messages.
>
> Regarding the other questions:
> - 1: Yes, limited by the message length |M|
> - 2: Yes, the feedback token is generated using a separate linear layer dedicated only for the feedback production.
> - 3: We do not intend to have the sender and receiver share the same vocabulary, as the idea is to create independent agents that only exchange information using the communication channel. As we are attempting to model human language evolution, we do not consider that a shared vocabulary is pre-existing between different agents.
> It is true that the sender’s messages in the feedback case depend on both the input object and the receiver’s feedback. We tried calculating topsim scores that also take into account the receivers messages (by concatenating the sender messages and the receiver messages) and found that these scores are even lower than the original topsim scores (e.g. model with $p_{noise}=0.7$, $(|A|, |V|) =  (4, 4)$: topsim_sender_receiver=0.15+-0.02 | topsim=0.19+-0.02).
> More recently, a number of works also started criticizing the use of topsim to measure compositionality in emergent communication [2,3]. Our work adds additional evidence for the weakness of this metric.
>
> [1] Graesser et al. 2020 Emergent Linguistic Phenomena in Multi-Agent Communication Games
>
> [2] Conclin & Smith 2023 Compositionality with Variation Reliably Emerges between Neural Networks
>
> [3] Korbak et al. 2020 Measuring non-trivial compositionality in emergent communication

---

### Official Review · Reviewer_R7xc · 2023-11-06

**Soundness:** 3 good
**Presentation:** 2 fair
**Contribution:** 2 fair
**Rating:** 5
**Confidence:** 3

**Summary:**

The paper explores the effects of conversational repair mechanisms on emergent communication in signaling games. The basic Lewis signaling game setup is extended to allow bidirectional communication through an interleaved feedback channel from receiver to sender. Noise is added to the communication channel by replacing tokens at random. Models trained with a feedback channel are found to achieve higher generalization performance under noisy conditions, even though the resulting languages are less compositional.

**Strengths:**

The experiments are interesting, testing multiple game configurations and noise levels. The code is open-sourced. Results are robust across settings and replicated in two different game paradigms. The paper clearly explains the methods, results, and implications. Allowing conversational repair is clearly an important step towards better models of human language evolution.

**Weaknesses:**

Methods: The most significant concern is that the apparent lack of compositionality (arguably the 'headline' result) is an artifact of the specific choice of how to inject noise (via random i.i.d. replacement with a special token). Modeling *misunderstanding*, uncertainty over *meaning* or *interpretation* at the message level, rather than uncertainty over the literal content of each message token, would better reflect the cases where repair arises in real-world communication.

Originality: While some details of the specific implementation here is novel, a number of prior works have also introduced interactive repair mechanisms into signaling games and are not discussed. Even the classic Steels (1995) simulations included a form of (binary) repair. Here are some salient examples from the more recent literature.

- van Arkel, Woensdregt, Dingemanse, & Blokpoel. (2022). A simple repair mechanism can alleviate computational demands of pragmatic reasoning: simulations and complexity analysis. CoNLL.
- Tria, Galantucci, & Loreto (2012). Naming a structured world: A cultural route to duality of patterning. Plos ONE.
- de Ruiter & Cummins. (2012). A model of intentional communication: AIRBUS (Asymmetric Intention Recognition with Bayesian Updating of Signals). SemDial.
- Silva & Roberts. (2016). Exploring the role of interaction in the emergence of linguistic structure. EVOLANG.
- White, Poesia, Hawkins, Sadigh, & Goodman. (2022). Open-domain clarification question generation without question examples. EMNLP.

Additional weaknesses:

* The feedback channel only allows binary signals. It's unclear how the results would be affected by a higher-dimensional space of feedback.
* Noise is only added to sender messages, but realistically noise would affect receiver feedback too, possibly shrinking the gap between the feedback vs. no-feedback models.
* No comparison to recent related work like van Arkel et al. is provided to situate the advances here.

In summary, while this work provides a good first investigation, the contributions are somewhat modest and incremental given prior exploration of feedback channels. It's difficult to know which aspects of the findings are general consequences of repair mechanisms, and which aspects are artifacts of specific choices about how noise is injected in this task setup. Addressing the above limitations would strengthen the novelty and significance of the study.

**Questions:**

* Typo: “defined by the number of symbols in the vocabulary V” — the vocabulary was previously denoted by X; V was defined was the set of possible values the attributes may take.

---

> ### Comment · Reviewer_3F5Q · 2023-11-16
> **Responding to above weaknesses**
>
> While I believe that this review brings up good points to consider about the
> paper, I think it over-weights the degree to which the potential weaknesses
> undermine the contributions.
>
> > Methods: The most significant concern is that the apparent lack of
> > compositionality (arguably the 'headline' result) is an artifact of the
> > specific choice of how to inject noise (via random i.i.d. replacement with
> > a special token).
>
> I do not think the lack of compositionality is or should be the headline
> result.  I believe that it is a good ancillary finding, but overall,
> compositionality---especially something as simple as toposim---is not an
> adequate measure of how "good" or "human like" or "important" an emergent
> language is (cf. [1] for issues with toposim directly and [2] for
> compositionality in EC more generally).  Thus, I would argue that it acceptable
> for the paper to make the observation it does without spending much time
> digging deeper.
>
> > Modeling misunderstanding, uncertainty over meaning or interpretation at the
> > message level, rather than uncertainty over the literal content of each
> > message token, would better reflect the cases where repair arises in
> > real-world communication.
>
> Uncertainty over meaning is certainly an interesting topic, but uncertainty
> over content is highly relevant to human communication (e.g., having
> a conversation at loud party, talking to someone who mumbles).
>
> > Originality: While some details of the specific implementation here is novel,
> > a number of prior works have also introduced interactive repair mechanisms
> > into signaling games and are not discussed.examples.
>
> The paper would definitely benefit from positioning itself explicitly vis-a-vis
> the paper mentioned in the review, but so far as I can tell, these paper are
> not specifically within the domain of "deep learning-based emergent
> communication" which has its own unique methods and concerns.  Therefore,
> I would argue that it is actually helpful to use a framework which has been
> studied before in closely related contexts (although, again, mentioning them
> and contextualizing itself are important here).
>
> ### References
>
> - [1] Korbak et al., 2020, https://arxiv.org/abs/2010.15058
> - [2] Kharitonov & Baroni, 2020, https://aclanthology.org/2020.blackboxnlp-1.2/

---

> ### Author Response · Authors · 2023-11-17
>
> Thanks for your thoughtful review!
>
> - We share the concern that the lack of compositionality could be an artifact of the specific implementation of noise. The goal of this project was to investigate the effect of conversational repair in the most basic setting (random replacement of message tokens with a noise token), which could e.g. translate to the real-world phenomenon of a listener not understanding a word because of some increased background noise. Indeed, there are many other possible situations in which conversational repair routines are employed, such as misunderstandings. The most basic implementation to model this phenomenon is to randomly permute a message token with another token from the vocabulary (instead of a special noise token). We ran the main experiments from the paper with this alternative noise implementation, and report the results in Section 4.1.3 of the updated version. The main conclusions hold: Models trained with feedback generalize better but produce less compositional languages. Future work should investigate the effect of even more noise implementations, such as combinations of non-understanding (noise token) and misunderstanding, as well as non-uniform distributions of noise.
>
> - Thanks for the very relevant references. We integrated them in the related work section of the updated manuscript and contextualize our work respectively. Our main contribution in comparison to these works is that we explicitly investigate the effect of a feedback channel by directly comparing models with and without feedback. Further, we employ deep-learning based modeling which scales to more realistic input, instead of only small-scale toy-language setups. This enabled us to perform the experiments described in Section 4.3.
>
> - It is true that the nature of the feedback will most likely change if the dimensionality of the feedback channel is increased. Instead of simple open clarification requests and/or acknowledgements, the model could leverage mechanisms such as restricted clarification requests or restricted offers. However, as the vocabulary size of the speaker is currently set to 2, we don’t think that an increase in the feedback channel dimensionality is actually useful in our setup. (We verify this claim by running a model with a feedback vocabulary size of 10, and find that it does not improve generalization performance: For a noise level of 0.9, the test accuracy drops from 0.75 (stddev: 0.07) to 0.70 (stddev: 0.07). As this slight drop is within the standard deviation, we assume it's caused by effects of the random initialization. Future work should investigate models with higher vocabulary sizes combined with increased feedback channel sizes.
>
> - Regarding noise on the feedback messages from the receiver: Yes, this has been discussed as a meaningful extension in Section 5, and will definitely be subject of future work.
>
> - Thanks as well for spotting the typo, this has been fixed.

---

> > ### Comment · Reviewer_R7xc · 2023-11-17
> >
> > Thank you for the thoughtful response, and the additional experiments. I still believe this work is somewhat 'undercooked' compared to other ICLR papers, but I will increase my score by a few points to reflect the strength of the revisions.

---

### Meta-Review · Area_Chair_qNCZ · 2023-12-06

**Metareview:**

The reviewers appreciate the experiments and results, noting that they are robust across conditions, and also appreciate the clarity of presentation.

However, concerns are raised regarding the modeling approach, novelty, generalizability, and further analysis needed. Not all reviewers agree with the weaknesses and the authors respond to the comments convincing some of the reviewers.

I would recommend borderline acceptance for this paper mainly given 3F5Q's support, but I am not an expert in this field.

**Justification For Why Not Higher Score:**

Only one reviewer is clearly supporting to accept this paper.

**Justification For Why Not Lower Score:**

One reviewer advocating for rejection was not convinced with the responses and another was borderline. This may signal that this paper needs more work.

---

### Decision · Program_Chairs · 2024-01-16

Accept (poster)